# Analysis of Mercury in Skin Lightening Cream by Microwave Plasma Atomic Emission Spectroscopy (MP-AES)

**DOI:** 10.3390/molecules26113130

**Published:** 2021-05-24

**Authors:** Hardoko I. Qudus, Purwadi Purwadi, Iis Holilah, Sutopo Hadi

**Affiliations:** 1Department of Chemistry, Faculty of Mathematics and Natural Sciences, Universitas Lampung, Bandar Lampung 35145, Indonesia; 2Indonesia National Agency of Food and Drug Control, Bandar Lampung 35228, Indonesia; mutubpomlpg@gmail.com; 3Public Senior High School 16, Bandar Lampung 35153, Indonesia; naisyaparis@gmail.com

**Keywords:** analytical method validation, mercury, MP-AES, skin lightening cream

## Abstract

This research aimed at developing an analysis method, which was optimized and validated to determine the content of mercury in skin lightening cream discovered in the market in Bandar Lampung, Indonesia, through the use of microwave plasma atomic emission spectroscopy (MP-AES). The optimization on the analysis method was conducted on pump rate, viewing position, and reductant concentration in order to obtain the highest mercury emission intensity, while the solution stability was optimized to know the stability of mercury in the solution. The result showed that the method developed had precision with a relative standard deviation of 2.67%, recovery value of 92.78%, and linearity with an r value of 0.993, respectively. The sensitivity of the instrument detection had a limit of analysis method detection and quantification of 0.59 and 1.98 µg/L, respectively. The results of the test of the lightening cream (8 of 16 samples) positively contained mercury in the range of 422.61–44,960.79 ng/g. Therefore the method of analysis developed may be used for routine analysis of chemicals in any cosmetics products.

## 1. Introduction

Cosmetics are materials intended for the use on the outer human body parts such as epidermis, hair, nail, lips, outside genital organs, teeth, and on mouth mucosa membrane, as they are specifically used for cleaning up, perfuming, performance altering, and body smell fixing while protecting and accurately maintaining the body condition [1].

One of the cosmetics often used by women is skin lightening cream. Unfortunately, most of these products in the market use mercury as an added material [1,2,3,4,5,6]. This is due to the fact that mercury has the ability to whiten skin color within a relatively short time. However, this metal accumulates in the human body, especially in the kidney, liver, and brain, which in turn causes negative effects on health [2,6].

Due to the high danger of mercury in cosmetics, this metal obtained high attention from the WHO, based on the adverse effect on human health [7]. A high concentration of mercury present in inorganic and organic compounds causes permanent toxicity in the brain, kidney, and embryo growth [3,4,6,7,8]. The poisonous level of mercury depends on its chemical form, with the lowest and highest appearances being ionic and metallic organomercury [1,2,7].

Mercury is a toxic metal and non-essential in the human body. The environmental problems are mostly caused by exposure to the organic mercury compound [1,7]. The inorganic metals are also converted to organic forms by the decomposition process of sulfate bacteria, which produces methylmercury, one of the most toxic forms of mercury compounds that are easily absorbed by cell membranes. Further, methylmercury causes problems in the nervous system, resulting in an abnormal function of the nervous system. Chronic toxicity due to mercury includes paresthesia, neuropathy perifer, cerebellar ataxia, akathisia, spasticity, memory loss, dementia, limited sight, dysarthria, hearing problems, a decrease in smell and taste values, tremor, and depression [8].

The determination of metals or metallic pollutants is generally performed by using flame atomic absorption spectroscopy (F-AAS) [5], hydride generation AAS [9,10,11], cold vapor AAS [12], inductively coupled plasma mass spectrometry (ICP-MS) [13,14,15], and, recently, microwave plasma atomic emission spectrometry (MP-AES) has also been utilized [14,15]. However, the operation of some of these techniques is high in cost, for example, F-AAS uses acetylene or nitrogen oxide as burning gas with a high flow rate speed, while also using a hollow cathode lamp with a certain lifetime. In addition to that, the detection limit of F-AAS is quite high, resulting in it not being good for metal contamination. Further, ICP-MS has a good detection limit, even though the use of argon limits its uses [13].

Karlson et al. [15] compared both MP-AES and ICP-MS methods, which included low and high cost analyses with detection limits in the units of µg/L and ng/L, respectively, in some metals such as Mg, Ca, Fe, and Al obtained from sun flowers. The results obtained were not significantly different, although few metals were not detected using MP-AES.

The development of the MP-AES analysis technique began with polluted metal on a variety of matrices, as reported by Wu et al. [16] and Yang et al. [17], who analyzed the material of Chinese herbals. Further, Zhao et al. [13] determined the content of some metals, including mercury, in skin and hair. The major challenges on the analysis of mercury in cosmetics include that the organic matrices are difficult to be digested, the wide variation of mercury concentration in the cosmetics products, and the composition of mercury is not clearly mentioned on the label [18,19].

MP-AES is a relatively new technique for metal analysis, which is also a metalloid in a variety of sample matrices. However, the use of this technique has not been supported with a developed analysis method, as no analytical development to analyze mercury in cosmetic cream has been performed. In this paper, the method of analysis using MP-AES to determine the content of mercury in some selected lightening creams obtained in the market in Bandar Lampung, Indonesia, was reported.

## 2. Results

The optimization of the nebulizer pressure and plasma viewing position was performed using MP-Expert software. The optimum pressure of the nebulizer was 160 kPa, with a viewing position at the zero point. Moreover, optimization was also conducted for every measurement. Using the optimum condition of the pressure nebulizer and viewing position the optimization of the pump speed was collected, in order to know the velocity at which the highest emission intention was obtained. Variations were also performed at 10, 20, 30, 40, and 50 rpm, as the optimum conditions were further obtained at 40 rpm, as shown in Figure 1.

The intensity of the measurement was directly proportional to the pump speed up to 40 rpm; this then dropped off at 50 rpm. However, the large pump speed resulted in a larger volume of the required test sample.

Subsequent optimization of the reductant concentration of NaBH_4_ was carried out at a pump speed of 40 rpm, which was conducted at variations of 0.5, 1.5, 2.0, and 2.5%, with optimum conditions at 1.5% of NaBH_4_. The results are shown in Figure 2.

Figure 3 showed the stability test using a standard mercury solution of 150 µg/L, the mercury solution in nitric acid was unstable in the first 2 h as shown by the intensity decrease of about 12.5%; however, this solution was stable up to 7 h, but the measurement after 7 h was not carried out. This change did not affect the measurement of the samples much as long as the measurement was conducted at an interval time of 2–7 h after the initial dissolving of the mercury standard and the samples.

Further, the precision test result had a very good repetition, which was used with an RSD value of 2.67%, as shown in Table 1.

The accuracy of the method was tested by a recovery test with the addition of mercury standard solution to the pure cream sample. The result of the accuration test method showed very high accuracy where the recovery percentage was satisfied, as shown in Table 2. The levels obtained were then calculated for the added cream sample with negative mercury, using the calibration curve equation of the standard mercury measurement of 5–90 µg/L, as indicated in y = 58.78x − 147.9.

The results of the linearity test method also showed that it was quite linear as it produced a regression equation of y = 46.87x + 277.75 with a coefficient correlation (r) of 0.993, as shown in Figure 4. The test was performed on a sample of lightening creams, which were added to mercury.

Furthermore, the method of the sensitivity test was conducted by using a serial standard mercury solution on a low concentration at 1–6 µg/L (Figure 5). This method was confirmed to be very sensitive, with LoD and LoQ at 0.59 and 1.98 µg/L, respectively.

Based on the use of the methods developed, the measurement result of the mercury content present in the sample skin lightening cream produced the following data shown in Table 3.

## 3. Discussion

MP-AES is a relatively new analytical technique, which was developed by Agilent Technologies [16]. This analysis technique offered several advantages when juxtaposed with older analytical methods, including AAS and ICP. Some of the advantages of MP-AES include low operating costs and safe usage. However, this analytical technique also revealed challenges in regard to the lack of available methods for the analysis of metals and semi-metallic elements on various matrices.

In this study, a mercury analysis was determined by MP-AES with the reaction chamber of MSIS (multimode sample introduction system), as well as the optimization results of the tool obtained by an optimum nebulizer pressure of 160 kPa, viewing position at the zero point, and pump rate of 40 rpm. The optimization of the tool was intended to obtain maximum measurement intensity and good repeatability. Further, the gas used to produce plasma was nitrogen, which was the in-situ product of the N-generator. The optimum pressure also produced maximum nebula, which was an aerosol filled with analytes. However, the view position was intended to obtain the highest intensity, which was captured by the detector.

At the optimization stage of NaBH_4_ (Figure 2), the optimum concentration value obtained was 1.5%. The addition of this concentration aimed to convert the mercury ion into mercuric hydrides, as it became more volatile and was easily delivered to the torch after nebulization. NaBH_4_ was also provided in excess at this stage of the process, as it intended to change the mercury analyte perfectly both in the reference standard solution and the sample.

According to Robbins and Caruso [9], the reaction that occurred between the reductant NaBH_4_ with mercury, arsenic, and selenium, was shown in Equations (1) and (2), where E was the metal Hg, As, and Se, with the values of m and n being similar, even though they were also likely to be different.
NaBH_4_ + 3H_2_O + HCl→H_3_BO_3_ + NaCl + 8 H^−^(1)
E^m+^+ H^−^ (excess)→EH_n_ + H_2_ (excess)(2)

Furthermore, the solution stability test (Figure 3) showed that the concentration was stable after 2 h, and this solution was relatively stable up to 7 h. This time span was required to be known when considering the analyst’s testing habits from preparation to completion of the measurements.

The validity of this method also fulfilled the requirements with the results of the precision test, which produced a relative standard deviation of 2.67% (Table 2). Further, the requirement for analytes at a level of 10 ng/mL was 21% [20]. Moreover, the accuracy of this method was 92.78%, which fulfilled the requirements for 10 ppb of recovery, at 60–115% [20]. This accuracy test was the most important considering the fact that mercury analytes were volatile at high temperatures. However, this method used the digestion technique at high temperatures and pressures (Table 4).

This analysis method also had good linearity, as reflected in the correlation coefficient (r) of 0.991 (Figure 5). Further, the sensitivity of the method was quite good, as shown from the LoD and LoQ (limit of detection and quantification) values of 15.01 and 50.02 ng/g, respectively. The measurement of mercury used a cold-vapor-AAS-produced detection limit of 0.02 ppm (µg/g) [21], while, when using ICP-MS, the detection limit obtained was 0.02 µg/L [22]. The obtained result in this work was 0.59 ng/g. Therefore, the test results of the lightening cream cosmetic material showed that in 8 of the 16 samples analyzed indicated mercury levels within the range of 422.61–44,960.79 µg/g. The method of analysis for mercury obtained in this work can be used and proposed for routine analysis of mercury in the cosmetics samples.

## 4. Materials and Method

### 4.1. Materials

All reagents used were of AR grade. These include sodium borohydride (NaBH_4_, (Merck, Kenilworth, NJ, USA), sodium hydroxide (NaOH, (Merck)), 37% hydrochloric acid (HCl, (Merck)), 30% hydrogen peroxide (H_2_O_2_, (Merck)), 65% nitric acid (HNO_3_, (Merck)), and deionized water from a Millipore purifier (Merck) with a resistance of 18.2 MΩ.cm. Additionally, the standardized reference using mercury ICP (Merck), traceable to NIST Standard Reference Material^®^ (SRM) Hg(NO_3_)_2_ in 10% nitric acid, equaled 1000 mg/L Hg.

The samples for method validation through cosmetic materials of the lightening creams selected were not detected using optimization results. Moreover, 16 test samples were obtained from some distribution area in Bandar Lampung.

### 4.2. Reagent Preparation

The nitric acid solution used for the standard solution and the digested sample was prepared by diluting 10 mL of nitric acid P.A. with deionized water, up to a volume of 1 L. Afterward, the reductant solution of sodium borohydride was gradually prepared by mixing 1.0% NaOH solution into 0.5–2.5 g of NaBH_4_ until a volume of 100 mL was reached [23]. Further, a 100 µL aliquot was obtained from the preparation of the standard mercury solution of 1000 mg/L and diluted with solvent HNO_3_ to a volume of 100 mL (Hg concentration of 1.0 mg/L). Afterward, the mercury series standard solution was further prepared, by serially pipetting 0.1–15.0 mL of the main standardized content, as each was dissolved with 100 mL of solvent in order to obtain a metallic concentration of 1–150 µg/L.

### 4.3. Measurement Optimization

#### 4.3.1. Initial Conditions of MP-AES System

Initial conditions of the MP-AES system are as follow:Reaction chamber: MSIS (multimode sample introduction system),Reductant: solution of sodium borohydride (NaBH_4_) 1.0%,Wavelength: 253.652 nm,Stabilization time: 15 s,Uptake time: 10 s,Pump speed: 50 rpm,Injection system: manual.

The optimization of the nebulizer pressure, viewing position, and pump speed was performed in an initial system using a standard solution of mercury at 150 µg/L. The optimization was then carried out through the use of MP-Expert software (MP-4100, Agilent, Santa Clara, CA, USA).

#### 4.3.2. Optimization of NaBH_4_ Reductant Concentration

The optimization of NaBH_4_ reductant concentration was also performed in an optimum condition, based on the previous experiments, by varying concentrations of 0.5–2.5%.

#### 4.3.3. Test of Solution Stability

The test of solution stability was measured with MP-AES at the optimum condition obtained from previous experiments using a standard mercury concentration of 150 µg/L. The measurement was performed periodically from 0–7 h starting from the preparation of the solution stock at 1 h.

### 4.4. Sample Preparation

The cream sample of 0.2–0.25 g was placed on a Teflon vessel, as 8 and 2 mL of nitric acid (concentrated) and H_2_O_2_ were added and left for 20 min, respectively. Afterward, this was digested in a microwave digestion oven at the conditions shown in Table 4.

After being digested, the samples were dissolved with deionized water in a volumetric flask of 50 mL, which was equipped with Whatman^®^ Filter Paper No. 1, as the solution materials were then measured with MP-AES. If the intensity was still too high, they were further diluted.

### 4.5. Precision Test

During this test, the cream sample of 0.2–0.25 g was placed on the Teflon vessel, as 200 µL stock solution of 10 mg/L mercury was added, with the measurement further carried out by using MP-AES. The precision test was performed six times, as the relative standard deviation was then calculated.

### 4.6. Accuracy Test

Using the result obtained with the precision test, the calculations of the recovery percentage and sample content were used with the calibration curve, which was obtained through the use of mercury concentration at 5–90 µg/L.

### 4.7. Linearity Test

A sample of 0.2–0.25 g was further placed in the vessel, and a stock mercury solution of 10 mg/L was added in the amount of 0, 100, 200, 300, 400, 500, and 600 µL. The measurement was then carried out using MP-AES, as the regression linear analysis was formed with the correlation coefficient (r) also being measured.

### 4.8. Test of Limit of Detection (LoD) and Limit of Quantity (LoQ)

The measurement of mercury solution stock with a concentration of 1–6 µg/L was performed, and the calculations of LoD and LoQ were 3 and 10 Sy(x)/b, respectively.

## 5. Conclusions

The result of the development analysis method of mercury on skin lightening cream through the use of MP-AES with the optimum condition of the MSIS (multimode sample introduction system) showed a wavelength with stabilization and uptake durations at 253.652 nm with 15 and 10 s, respectively. Further, the introduction system, in which the manual was obtained, was the nebulizer pressure of 160 kPa, viewing position at point zero, pump rate at 40 rpm, and NaBH_4_ concentration at 1.5%, with a duration of the test solution stability at 7 h. Furthermore, the validation method fulfilled the requirements of the precision test, with a relative standard deviation of 2.67%. The accuracy of this method, based on the recovery test, was also 92.78%, with the linearity of the correlation coefficient at 0.993. Further, the limits of the instrumental detection and quantitation obtained were 0.59 and 1.98 µg/L, respectively. Moreover, the analysis method results of LoD and LoQ were 15.01 and 50.02 ng/g, respectively. Therefore, out of the 16 cosmetic samples of skin lightening creams tested, 8 were detected to contain mercury, with a content range of 422.61–44,960.79 µg/g. Therefore, in future work, it is suggested to use other solvents such as HCl or BrCl, and other equivalent techniques such as CV AAS or ICP-MS to compare the results obtained.

## Figures and Tables

**Figure 1 molecules-26-03130-f001:**
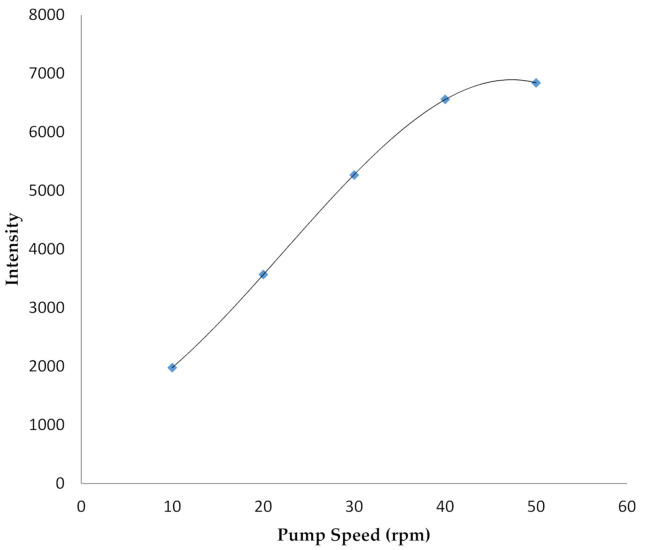
The effect of varying pump speed on the intensity of mercury emission at 253.652 nm.

**Figure 2 molecules-26-03130-f002:**
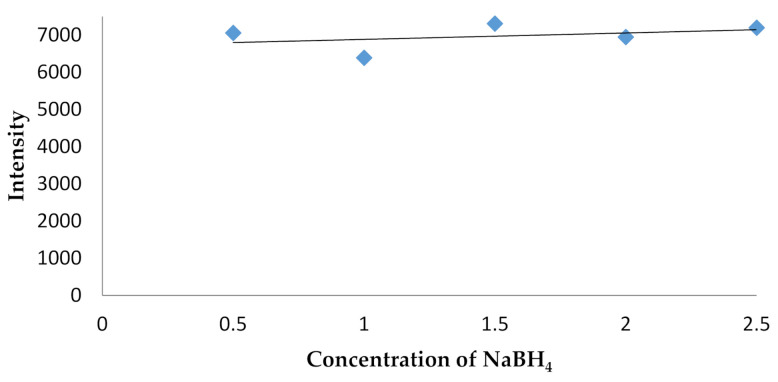
Optimization of NaBH_4_ concentration.

**Figure 3 molecules-26-03130-f003:**
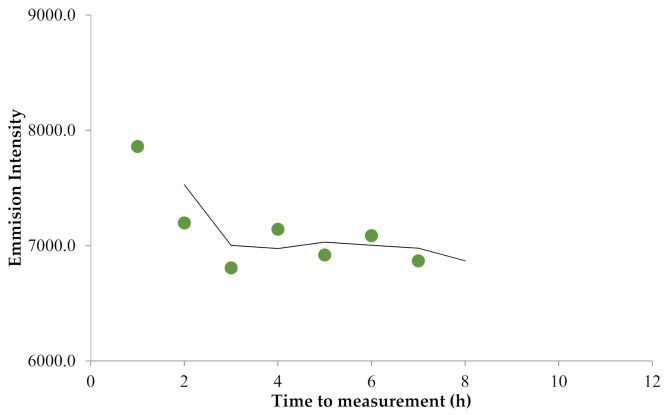
Stability test of the solution.

**Figure 4 molecules-26-03130-f004:**
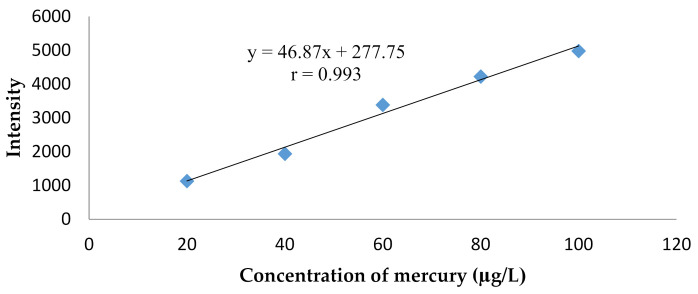
The result of linearity test method.

**Figure 5 molecules-26-03130-f005:**
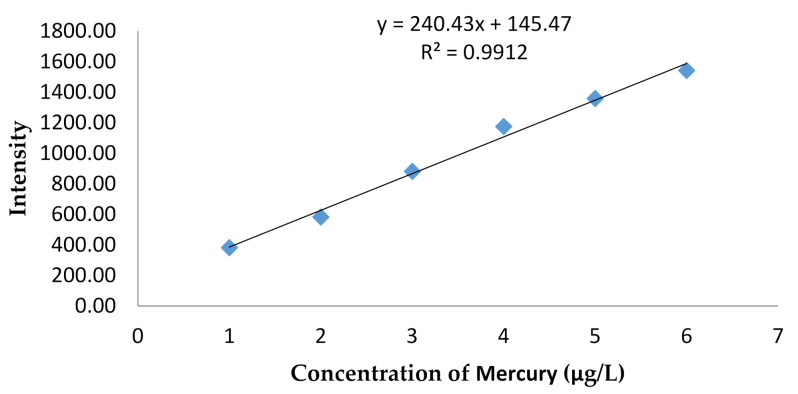
Calibration curve of mercury on low concentration.

**Table 1 molecules-26-03130-t001:** Precision test of the method used.

Label	Intensity
Precision 1	2118.279
Precision 2	2071.708
Precision 3	2040.439
Precision 4	1993.080
Precision 5	2005.446
Precision 6	1973.390
Average	2033.724
Standard deviation (SD)	54.28
RSD (%)	2.67

**Table 2 molecules-26-03130-t002:** Accuration test of the method.

Label	Intensity	Expected Value (ng/g)	Obtained Value (ng/g)	Recovery (%)
Recovery 1	2118.279	40.00	38.55	96.37
Recovery 2	2071.708	40.00	37.76	94.39
Recovery 3	2040.439	40.00	37.23	93.06
Recovery 4	1993.080	40.00	36.42	91.05
Recovery 5	2005.446	40.00	36.63	91.57
Recovery 6	1973.390	40.00	36.08	90.21
		Average		92.78
		The smallest		90.21
		The biggest		96.37

**Table 3 molecules-26-03130-t003:** Mercury concentration in the sample measured.

Sample Code	Mercury Content (µg/g)
1	n.d.
2	n.d.
3	n.d.
4	n.d.
5	1901.82
6	n.d.
7	422.61
8	n.d.
9	n.d.
10	n.d.
11	836.27
12	44,960.79
13	26,212.56
14	6681.45
15	4928.30
16	4360.44

n.d. = not detected.

**Table 4 molecules-26-03130-t004:** The condition of the sampling technique in the microwave digestion oven.

Step	Temperature (°C)	Time (min)	Power (Watt)
1	130	10	800
2	160	10	800
3	190	10	800

## Data Availability

The data produced from this study can be requested from the corresponding author upon reasonable request.

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
