# Peer review of "Analysis of Mercury in Skin Lightening Cream by Microwave Plasma Atomic Emission Spectroscopy (MP-AES)"

_molecules, 2021, doi:10.3390/molecules26113130_

Round 1

Reviewer 1 Report

This paper unfortunately needs extensive English revision, I know it's hard to write in a second language, but the text is not up to the standard expected of a scientific journal.

My main concern is that the authors haven't stabilised the mercury in solution, and haven't realised that the solution isn't stable. Their figure 3 clearly demonstrates that the standard solution isn't stable over two hours.

I have made extensive note and annotations on the pdf file of the paper to help the authors. The data the report are important and should be published. The paper needs to be brought up to publication quality though. It is disorganised in places and uses incorrect terms and concept. For instance the authors report 2 detection limits. One lower than the other. Probably one is the method detection limit, but we can't tell as the units are the same.

Author Response

We would like to thank you for your very valuable comments that really help us in doing the revision.

  1. The manuscript has actually been proofread by a native, however since you commented that the language is still hard to understand, we will do another proofreading with MDPI language editing service upon your agreement with the revision which has been done to the manuscript.
  2. We have attempted correction through the manuscript based on your comments.
  3. The typo errors have been corrected.
  4. The measurement unit has been changed µg/L.
  5. According to Reference no. 21, the NaBH4 was stabilized using alkaline solution.
  6. so this is the instrumental detection limit? Yes, it is
  7. how is the LOD 15 ng/ml when they managed a calibration between 1 and 6 ppb (µg/L) or ng/mL? Is this th method detection limit in face cream? The authors have two detection limits, they should specifiy which is which. If this is the method detection limit it should be in ug/g. è The detection limit of the instrument used was 0.59 ng/mL (µg/L) which was obtained with calibration curve between 1 and 6 ppb (µg/L) or ng/ml as in Figure 5.
  8. memory effects at high concentrations? à I am not sure but may be not, because we were the first people that used the instrument (MP-AES) to measure the mercury.
  9. The authors don't report if they used a standard spray chamber or a gas liquid separator è We did mention in the 4.3.1 that we used MSIS (multimode sample introduction system).
  10. in water, not in dilute HNO3 I suppose. We diluted NaBH4 with 1% NaOH.
  11. 0.1% HNO3 is not enough to keep mercury stable in solution. You normally need a stablising agent, either HCl or gold solution or BrCl à Based on the result of experiment as shown in Figure 3, the mercury solution is stable after the Second hour, and stable up to 7 h. The most important thing, in the preparation of the mercury sample and the standard solution must be prepared at the same time.
  12. are these the optimised conditions? No, this step is an attempt to find the optimum condition which then using MP-Expert Software to optimise the maximum wavelength, nebulizer pressure, viewing position, and pump speed. After the optimum condition for measurement obtained, the concentration of NaBH4 reductant was measured.

Reviewer 2 Report

  1. What does this sentence "The optimized analysis method was performed on pump rate, viewing position, reductant concentration, and solution stability" mean?
  2.  This sentence is not clear, please rephrase it. "The result showed that the method developed had precision with relative standard deviation, recovery value, and linearity with r, at 2.67%, 92.78%, and 0.993, respectively."
  3. What do authors mean by "the limits of analysis method detection and quantification"
  4. In this sentence "However, the use of this technique has not been supported with ample analysis method,..." what do authors mean by ample analysis?
  5. The authors need to improve the introduction by discussion the majors challenges encountered when one analyze mercury in cosmetics.
  6. To enhance the novelty of this study, authors must compare univariate and multivariate optimization techniques. It would be interesting to see how will the differ or agree. Also it would interesting identify parameters and interactions that influence better performance of the method.
  7. What informed the factors that were selected and optimized?
  8. There is something wrong with Table 2. Table 2 was supposed to be Table 1. what is the relevance of this table?
  9. "..where the recovery percentage connected.." What the phrase mean?
  10. "The levels obtained were then calculated in the added sample,
     using the calibration curve equation of the standard mercury measurement of 5 - 90 ng/mL, as indicated in y = 58.78x - 147.9" in this sentence, the sample was added to what?
  11. The results obtained must be validated using an ICP-OES or CV-AAS or mercury analyser or ICP-MS.
  12. English in this manuscript need some serious improvement. It is difficult to understand the content of the paper. 

Author Response

We would like to thank you for your very valuable comments that really help us in doing the revision.

  1. The manuscript has actually been proofread by a native, however since you commented that the language is still hard to understand, we will do another proofreading with MDPI language editing service upon your agreement with the revision which has been done to the manuscript.
  2. We have attempted correction through the manuscript based on your comments.
  3. The typo errors have been corrected.
  4. The measurement unit has been changed µg/L.
  5. The optimized analysis method was performed on pump rate, viewing position, reductant concentration, and solution stability è we have corrected the optimization on the analysis method was conducted on pump rate, viewing position, and reductant concentration in order to obtain the highest mercury emission intensity, while solution stability was optimized to know the stability of mercury in the solution
  6. "The result showed that the method developed had precision with relative standard deviation, recovery value, and linearity with r, at 2.67%, 92.78%, and 0.993, respectively." We have rephrased it
  7. The limits of analysis method detection and quantification è we have corrected as The sensitivity of the instrument detection has limit of analysis method detection and quantification of 0.59 and 1.98 µg/L, respectively.
  8. The authors need to improve the introduction by discussion the majors challenges encountered when one analyze mercury in cosmetics è we have added some majors challenges in the introduction.
  9. To enhance the novelty of this study, authors must compare univariate and multivariate optimization techniques. It would be interesting to see how will the differ or agree. Also it would interesting identify parameters and interactions that influence better performance of the method è As the experiment has been concluded a while ago, it is hard to provide such data. This work was measured the trace element thus in order to know the better technique used, we compared our result with other results obtained by other technique analysis.
  10. What informed the factors that were selected and optimized? è It has been mentioned in Conclusion
  11. There is something wrong with Table 2. Table 2 was supposed to be Table 1. what is the relevance of this table? à it has been corrected
  12. "..where the recovery percentage connected.." What the phrase mean? à à it has been corrected
  13. "The levels obtained were then calculated in the added sample, using the calibration curve equation of the standard mercury measurement of 5 - 90 ng/mL, as indicated in y = 58.78x - 147.9" in this sentence, the sample was added to what? è We mean that the added cream sample with negative mercury
  14. The results obtained must be validated using an ICP-OES or CV-AAS or mercury analyser or ICP-MS è Unfortunately, we do not have such instruments for comparison, This method is proposed as the solution for official analysis method for mercury in cosmetics.
  15. English in this manuscript need some serious improvement. It is difficult to understand the content of the paper è we will do proofreading again after the final approval.

Round 2

Reviewer 1 Report

The science is fine, although I'd like the authors to address the fact that the mercury standard loses intensity for the first two hours, then stabilises at a lower unknown concentration. They should admit this and justify it and demonstrate that they're results aren't changed because of it. Are all the standards affected by the same percentage drop and are they used after the same time period?

The paper needs extensive English editing and correcting.

Author Response

Thank you for your help in commenting this statement.

We have corrected as follows:

Figure 3 showed the stability test using a standard mercury solution of 150 µg/L, the mercury solution in nitric acid was unstable in the first 2 h as shown by intensity decrease of about 12.5%, however this solution was stable up to 7 h, but the measurement after 7 h was not carried out. This change did not much affect the measurement of the samples as long as the measurement was conducted at interval time of 2-7 h after the initial dissolving of the mercury standard and the samples.

We will do the proofreading once our response has been approved.

Reviewer 2 Report

The authors tried to responds to the comments however, some responses are not convincing. e.g. As the experiment has been concluded a while ago, it is hard to provide such data. This work was measured the trace element thus in order to know the better technique used, we compared our result with other results obtained by other technique analysis. If the work was done a while ago, does it means it cannot be now. Does this response means, the work done is not reproducible?

Author Response

Thank you for your nice and precious comment.

This work surely is reproducible. I mean since we do not have other instruments to compare the work that we have done, then we rather compare with other data obtained from literatures.

It is why we mention the result of mercury measurement using other techniques (line). We have also added suggestion for our future work although with some limitation of facilities we have. I mean I have to do it at other institution.

We will do the proofreading once our response has been approved.